

# On the tear proteome of the house mouse (*Mus musculus musculus*) in relation to chemical signalling

Romana Stopkova, Petr Klempt, Barbora Kuntova and Pavel Stopka

BIOCEV group, Department of Zoology, Faculty of Science, Charles University, Prague, Czech Republic

## ABSTRACT

Mammalian tears are produced by lacrimal glands to protect eyes and may function in chemical communication and immunity. Recent studies on the house mouse chemical signalling revealed that major urinary proteins (MUPs) are not individually unique in *Mus musculus musculus*. This fact stimulated us to look for other sexually dimorphic proteins that may—in combination with MUPs—contribute to a pool of chemical signals in tears. MUPs and other lipocalins including odorant binding proteins (OBPs) have the capacity to selectively transport volatile organic compounds (VOCs) in their eight-stranded beta barrel, thus we have generated the tear proteome of the house mouse to detect a wider pool of proteins that may be involved in chemical signalling. We have detected significant male-biased (7.8%) and female-biased (7%) proteins in tears. Those proteins that showed the most elevated sexual dimorphisms were highly expressed and belong to MUP, OBP, ESP (i.e., exocrine gland-secreted peptides), and SCGB/ABP (i.e., secretoglobin) families. Thus, tears may have the potential to elicit sex-specific signals in combination by different proteins. Some tear lipocalins are not sexually dimorphic—with MUP20/darcin and OBP6 being good examples—and because all proteins may flow with tears through nasolacrimal ducts to nasal and oral cavities we suggest that their roles are wider than originally thought. Also, we have also detected several sexually dimorphic bactericidal proteins, thus further supporting an idea that males and females may have adopted alternative strategies in controlling microbiota thus yielding different VOC profiles.

## INTRODUCTION

The genome of the mouse contains at least 55 genes for lipocalins and—due to their capacity to transport VOCs in their eight-stranded beta barrel—many of them are involved in chemical communication (*Logan, Marton & Stowers, 2008*; *Mudge et al., 2008*; *Sam et al., 2001*; *Sharrow et al., 2002*; *Stopková et al., 2009*; *Timm et al., 2001*; *Zidek et al., 1999*). MUP production is male-biased and (sub-)species-specific in the mouse urine (*Stopková et al., 2007*), and have strong effects on the reproductive success of the signaller (*Thonhauser et al., 2013*) as an honest, cheat-proof display of an individual's health and condition (*Zala et al., 2015*; *Zala, Potts & Penn, 2004*). Because they are male-biased, the signals that are transported by highly homologous and invariable MUPs (*Enk et al., 2016*; *Thoss et al., 2016*;

Corresponding author
Pavel Stopka, pstopka@natur.cuni.cz

*Thoß et al., 2015*), have the capacity to regulate reproductive behaviour of female receivers (*Janotova & Stopka, 2011*; *Ma, Miao & Novotny, 1999*; *Novotny et al., 1986*; *Stopka, Janotova & Heyrovsky, 2007*), and have been proposed to function in a variety of social interactions (*Hurst & Beynon, 2004*; *Hurst et al., 2001*; *Mucignat-Caretta & Caretta, 1999*; *Nelson et al., 2015*; *Roberts et al., 2010*; *Rusu et al., 2008*). Lipocalins are also abundant in tears, e.g., MUP4, MUP5, OBP5-7, LCN11 (*Shahan et al., 1987*; *Shahan, Gilmartin & Derman, 1987*; *Stopkova et al., 2016*) and their roles in chemical communication have been particularly suggested for OBPs, namely for hamster MSP (i.e., male-specific submandibular salivary gland protein) and FLP (i.e., female lacrimal protein) which are dominantly expressed by female lacrimal glands (*Srikantan & De, 2008*; *Srikantan, Parekh & De, 2005*) and are regulated by sex hormones (*Ranganathan & De, 1995*; *Ranganathan, Jana & De, 1999*; *Srikantan, Parekh & De, 2005*). Another hamster OBP similar to MSP and FLP is Aphrodisin. It was detected in hamster vaginal secretions and presumably elicits copulatory behaviour in males (*Macrides et al., 1984*; *Singer et al., 1986*), and is also present in the urine of mole rats (*Hagemeyer et al., 2011*). MSP, FLP, and Aphrodisin are phylogenetically close to mouse OBPs (*Stopkova et al., 2014*). Mammalian OBPs were thought to be coded only by a few genes per species (*Cavaggioni & Mucignat-Caretta, 2000*; *Nagnan-Le Meillour et al., 2014*; *Pes et al., 1992*; *Pes & Pelosi, 1995*). However, there are more genes for OBPs in the mouse genome (*Stopková et al., 2009*; *Stopkova et al., 2010*) and, thus, we provided alternative names based on their position on chromosome X as *Obp1*, *Obp2*, *Obp5* (synonym in C57BL—*Obp1a* (*Pes et al., 1998*)), *Obp6*, *Obp7* (synonym in C57BL—*Obp1b* (*Pes et al., 1998*)), and *Obp8*, where *Obp3-ps* and *Obp4-ps* are pseudogenes. OBPs share typical lipocalin motif GxW as well as a specific disulfide bond (Cys38–Cys42), which represents a strong OBP-diagnostic motif CXXXC found in many mammalian OBPs including Probasin, and which are encoded by genes on the mouse chromosome X (*Srikantan, Parekh & De, 2005*; *Stopkova et al., 2014*; *Stopková et al., 2009*). Mouse *Obp* genes form two phylogenetic sub-clusters: the group-A and the group-B *Obp*s. The ancestral group-A *Obp*s include *Obp1* and *Obp2* (and *Obp3-ps*) whilst the later evolved group-B *Obp*s include *Obp5, Obp6, Obp7*, and *Obp8* (and *Obp4-ps*) (*Stopkova et al., 2016*). Furthermore, *Msp* and *Flp* are orthologs to group-A *Obp*s, whilst *Aphrodisin* belongs to group-B *Obp*s (*Stopkova et al., 2010*).

In our recent study employing qPCR techniques, we have determined the mRNA expression sites for OBPs and several MUPs, and provided evidence that the exorbital lacrimal glands produce high quantities of mRNAs coding OBP5, OBP6, female-biased OBP7, and also male-biased MUP4, and non-dimorphic MUP5 and LCN11 (*Stopkova et al., 2016*). Interestingly, those lipocalins that are produced by lacrimal and nasal tissues, are finally transported to the oral cavity where they are detectable in saliva (*Stopka et al., 2016*). These proteins are also spread onto the fur with saliva during selfgrooming where they may function as chemical signals. In tears, the male-biased MUP4 is particularly important for its affinity to the male-derived pheromone 2-*sec*-butyl-4,5-dihydrothiazole—SBT (*Sharrow, Novotny & Stone, 2003*; *Sharrow et al., 2002*) which causes inter-male aggression and estrus synchrony (*Jemiolo, Harvey & Novotny, 1986*; *Novotny et al., 1985*). Male tears, however, also contain exocrine gland-secreted peptides of which ESP1 has been shown to

enhance female sexual behaviour through a specific vomeronasal receptor (*Kimoto et al., 2005*). Thus, the presence of MUP4 with its ligands and of ESP1 in the mouse tears and saliva (*Stopka et al., 2016*) may function as the signals that explain the observation of *Luo, Fee & Katz (2003)* and *Luo & Katz (2004)*, who reported that mouth and facial areas are the first and the most frequently investigated areas during mouse social contacts, which causes strong neuronal activity responses in accessory olfactory bulbs (*Luo, Fee & Katz, 2003*).

MUPs and OBPs may also be used as carriers of VOCs stemming out of metabolic and bacterial degradation, and of other potentially toxic waste in mice (*Kwak et al., 2011*; *Kwak et al., 2016*; *Larsen, Bergman & Klassonwehler, 1990*; *Petrak et al., 2007*) and in other taxa including humans (*Akerstrom et al., 2007*; *Lechner, Wojnar & Redl, 2001*), cows and pigs (*Grolli et al., 2006*), and elephants (*Lazar et al., 2002*). Thus, scavenging is seen as their parallel—and presumably ancestral—role within the 'Toxic waste hypothesis of evolution of chemical communication' (*Stopková et al., 2009*), which states that, the original function of lipocalins was to transport harmful chemicals out of the body or for their internalization in lysosomes (*Strotmann & Breer, 2011*). These compounds—and especially those that were sexually dimorphic—were an ideal source for natural selection during evolution of sexual signalling due to a link between the level of metabolic activity and individual quality.

Besides lipocalins, tears also contain antimicrobial proteins, which keep the exposed parts of the eyeball hostile to pathogens (*Walcott, 1998*; *Zoukhri, 2006*). For example, secretory IgA inhibits pathogen adhesion, phospholipase A2 hydrolyses phospholipids in bacterial membranes and various growth factors maintain cornea proliferation and regeneration, reviewed in *Fluckinger et al. (2004)*. Specific antimicrobial activity has been demonstrated for the mouse lipocalin LCN2, which is up-regulated as a response to inflammation in mucosal tissues (*Flo et al., 2004*; *Goetz et al., 2002*), and which scavenges for catecholate-type siderophores that bacteria use to sequester free iron (*Flo et al., 2004*). LCN2 is equally present in male and female saliva (*Stopka et al., 2016*). Some mechanisms of defence such as bactericidal proteins from the PLUNC (palate, lung, and nasal epithelium clone) protein family are sexually dimorphic, thus differentially defending the mucosal layers of the body against pathogenic microbiota. These include for example the bactericidal/permeability-increasing proteins - BPI (*Leclair, 2003a*; *LeClair, 2003b*) which are male-biased in the mouse saliva (*Stopka et al., 2016*). Thus, the products from defeated bacteria and from symbiotic microbiomes may be sexually dimorphic due to the sexually dimorphic expression of anti-microbial proteins. They may contribute to an existing pool of compounds that may be recognized as individual signals by which the mice recognize an individual's health (*Zala et al., 2015*; *Zala, Potts & Penn, 2004*). Chemodetection of such microorganism-associated molecular patterns (MAMPs) occurs at many places in the body including specific sets of chemosensory neurons in the mammalian nose (*Bufe & Zufall, 2016*).

The aim of this paper was to characterize the tear proteome from wild individuals of the house mouse (*M. m. musculus*). We focused on the detection of abundant and sexually-dimorphic proteins and especially on those that have the potential to transport sexual signals in their beta barrel (i.e., lipocalins) and those that may be involved in generating sex-specific VOC profiles, including e.g., antimicrobial peptides. This paper builds upon

our previous study where we identified several lipocalins across different orofacial tissues with qPCR (*Stopkova et al., 2016*), and upon our study on the saliva proteome (*Stopka et al., 2016*) where we demonstrated that many salivary proteins (e.g., LCNs, and OBPs) are not expressed by submandibular glands but are produced elsewhere in nasal and lacrimal glands/tissues. Here we further developed this aim with the state-of-the-art label-free LC-MS/MS techniques to provide further evidence on the house mouse tear protein content.

## MATERIALS AND METHODS

### Ethical standards

All animal procedures were carried out in strict accordance with the law of the Czech Republic paragraph 17 no. 246/1992 and the local ethics committee of the Faculty of Science, Charles University in Prague chaired by Dr. Stanislav Vybíral specifically approved this study in accordance with accreditation no. 27335/2013-17214 valid until 2019. Animals were sacrificed by cervical dislocation.

### Animals

Fourteen individuals of the house mouse (the eastern form, *M. m. musculus*) used in this study were captured in the Czech Republic near Bruntál - 49.9884447N, 17.4647019E (one male, one female), in Velké Bílovice - 48.8492886N, 16.8922736E (three males, three females), Prague-Bohnice - 50.1341539N, 14.4142189E (three males, three females). All animals were trapped in human houses and garden shelters. On the day of capture or the next day, all animals were transferred to our animal facility. Each animal was caged individually with *ad libitum* access to water and food.

### Tear collection

Eye lavage was used as a non-invasive method of tear collection. Both eyes were carefully rinsed with 10 µl of the saline physiology solution by a gentle pipetting and samples were then pooled. The process was repeated three times with at least a two hour interval between every rinsing, and each sample was analysed twice with MS to produce mean values from the methodology duplicates. This was done in the 'in-house' Mass Spectrometry and Proteomics Service Laboratory, Faculty of Science, Charles University in Prague.

### Protein digestion

Protein samples were precipitated with the ice-cold acetone and followed by a re-suspension of dried pellets in the digestion buffer (1% SDC, 100 mM TEAB—pH = 8.5). Protein concentration of each lysate was determined using the BCA assay kit (Fisher Scientific, Waltham, MA, USA). Cysteines in 20 µg of proteins were reduced with a final concentration of 5 mM TCEP (60 °C for 60 min) and blocked with 10 mM MMTS (i.e., S-methyl methanethiosulfonate, 10 min Room Temperature). Samples were cleaved with trypsin (i.e., 1/50, trypsin/protein) in 37 °C overnight. Peptides were desalted on a Michrom C18 column.

### nLC-MS$^2$ analysis

Nano Reversed phase columns were used (EASY-Spray column, 50 cm × 75 µm ID, PepMap C18, 2 µm particles, 100 Å pore size). Mobile phase buffer A was composed of

water, 2% acetonitrile and 0.1% formic acid. Mobile phase B contained 80% acetonitrile, and 0.1% formic acid. Samples were loaded onto a trap column (Acclaim PepMap300, C18, 5 µm, 300 Å Wide Pore, 300 µm × 5 mm, five Cartridges) for 4 min at 15 µl/min loading buffer was composed of water, 2% acetonitrile and 0.1% trifluoroacetic acid. After 4 min ventile was switched and Mobile phase B increased from 2% to 40% B at 60 min, 90% B at 61 min, hold for 8 min, and 2% B at 70 min, hold for 15 min until the end of run.

Eluting peptide cations were converted to gas-phase ions by electrospray ionization and analysed on a Thermo Orbitrap Fusion (Q-OT-qIT; Thermo Fisher, Waltham, MA, USA). Survey scans of peptide precursors from 400 to 1,600 m/z were performed at 120K resolution (at 200 m/z) with a $5 \times 10^5$ ion count target. Tandem MS was performed by isolation at 1.5 Th with the quadrupole, HCD fragmentation with normalized collision energy of 30, and rapid scan MS analysis in the ion trap. The $MS^2$ ion count target was set to $10^4$ and the max injection time was 35 ms. Only those precursors with charge state 2–6 were sampled for $MS^2$. The dynamic exclusion duration was set to 45 swith a 10 ppm tolerance around the selected precursor and its isotopes. Monoisotopic precursor selection was turned on. The instrument was run in top speed mode with 2 s cycles.

## Protein analysis

All data were analysed and quantified with MaxQuant software (version 1.5.3.8) (*Cox et al., 2014*). The false discovery rate (FDR) was set to 1% for both proteins and peptides and we specified a minimum peptide length of seven amino acids. The Andromeda search engine was used for the MS/MS spectra search against the Uniprot *Mus musculus* database (downloaded on June, 2015), containing 44,900 entries. Enzyme specificity was set as C-terminal to Arg and Lys, also allowing cleavage at proline bonds (*Rodriguez et al., 2008*) and a maximum of two missed cleavages. Dithiomethylation of cysteine was selected as fixed modification and N-terminal protein acetylation and methionine oxidation as variable modifications. The "match between runs" feature of MaxQuant was used to transfer identifications to other LC-MS/MS runs based on their masses and retention time (maximum deviation 0.7 min) and this was also used in all quantification experiments. Quantifications were performed with the label-free algorithms described recently (*Cox et al., 2014*) using a combination of unique and razor peptides. To detect differentially expressed / abundant proteins, we used the Power Law Global Error Model (PLGEM) (*Pavelka et al., 2004*) within the *Bioconductor package* in R software (*Gentleman et al., 2004*). This model was first developed to quantify microarray data (*Pavelka et al., 2004*); however, due to similar statistical properties—namely the distribution of signal values deviating from normality—it has proved to be an amenable model for the quantification of label-free MS-based proteomics data (*Pavelka et al., 2008*). Next, we calculated the signal-to-noise ratio—STN (equation provided in *Pavelka et al., 2008*), because it explicitly takes unequal variances into account and because it penalizes proteins that have higher variance in each class more than those proteins that have a high variance in one class and a low variance in another (*Pavelka et al., 2004*). PLGEM was fitted on a set of replicates from female data, thus setting experimental baseline. Correlation between the mean values and standard deviations was high ($r^2 = 0.96$, Pearson = 0.94) so we continued with the

resampled STNs and calculated differences with corresponding *p*-values between males and females.

## Protein surface modelling

The surface electrostatics modelling involved several steps. First, we downloaded the structures from the RSCB Protein Data Bank (http://www.rcsb.org/) under accession IDs: 3S26, 1I04 and 2L9C, respectively. Because the mouse OBP1 structure has no record in the database we had to predict it with i-TASSER (Iterative Threading ASSEmbly Refinement) program (http://zhanglab.ccmb.med.umich.edu/I-TASSER/) with the rat ortholog OBP1F (PDB ID: 3FIQ, 76% similarity) as template for the homologous modelling. Next, we used PyMOL - Molecular Graphic System (version 1.7.0.0) with APBS (Adaptive Poisson-Boltzman Solver) plugin to model the electrostatics with the default software settings.

## RNAseq: samples, cDNA, sequencing, and analysis

Individual mice were sacrificed next day after the last tear sampling. The exorbital lacrimal glands (i.e., one gland per individual, the other one is stored) were dissected and immediately placed into RLT buffer (Qiagen, Hilden, Germany) and homogenised in MagNALyser (Roche) for 30 s at 6,000 rpm. RNA was isolated using the RNeasy Mini Kit (Qiagen, Hilden, Germany) according to the manufactures protocol with on-column DNase I treatment. The purity and concentration of eluted RNA was measured with a NanoDrop ND1000. The quality of RNA was checked on agarose gel electrophoresis (AGE). RNA was stored at $-70\,^\circ$C pending further use. For the next step, we selected only high quality samples from four male and four female replicates.

cDNA was prepared using the SMARTer PCR cDNA Synthesis Kit (Clontech, Mountain View, CA USA) and amplified with Advantage 2 PCR Kit (Clontech, Mountain View, CA USA). Both procedures were handled according to protocol for Trimmer-2 Normalization Kit (Evrogen, Moscow, Russia). The products of optimized cDNA amplification were then loaded on AGE. For each sample, only the area of product in range from $\sim$400 bp to $\sim$1,300 bp (well visible area full of bands) was excized from the gel and the DNA products were extracted using the Gel/PCR DNA Fragments Extraction Kit (Geneaid, New Taipei City, Taiwan). Appropriate amounts of size-selected products were then secondarily amplified according to the recommended protocol from Evrogen. Products of secondary amplification were purified using MiniElute PCR Purification Kit (Qiagen, Hilden, Germany). Purified products (and the range where they emerge) were checked on AGE. Purity was analysed with NanoDrop ND1000. Concentration was measured/determined using Quant-it Pico Green dsDNA Assay Kit (Invitrogen, Carlsbad, CA, USA) and fluorimeter (Hoefer DQ 300). Rapid Library (RL) was prepared for each transcriptome (four males and four females) according to Rapid Library Preparation Manual (my454.com). Equal amounts from each of eight Rapid Libraries ($10^7$ molecules per µl dilution) were mixed and then used for emPCR.

We conducted 454 RNA-sequencing with a desktop pyro-sequencer GS Junior from Roche using the long reads mode. To increase the precision of transcript mapping, we excised from a gel and sequenced only transcripts between $\sim$400 and 1,300 bp. Transcripts of this length include those of genes, described for their involvement in chemical

communication (e.g., lipocalins). This method is amenable to further analyses because the nebulization step is skipped and, therefore, whole transcripts instead of their fragments are further pyro-sequenced and mapped. We estimated particular expression levels from the number of uniquely mapped transcripts assigned to each annotated gene. All steps followed the provider's instructions for sequencing with GS Junior (emPCR Amplification Method Manual Lib-L and Sequencing Method Manual; Roche, Basel, Switzerland). We obtained >165,000 high quality (HQ) reads. HQ 454 Reads were multiplexed, trimmed (i.e., using a trimming database that contains primers used for library preparations), filtered and aligned into contigs against *Mus musculus* cDNA database (''the super-set of all known, novel and pseudo gene predictions''; ensembl.org, 17-FEB-2015 version) and using GS Reference Mapper (Roche, Basel, Switzerland). Differential expression was analysed in R software using the *DEseq* routine within the *Bioconductor package* (*Gentleman et al., 2004*).

### RNA-seq data availability

The transcriptome data is provided as bam files in 'Sequencing Read Archive' (www.ncbi.nlm.nih.gov/sra) under the accession numbers SRP063762 and BioProject PRJNA295909.

## RESULTS

### The tear proteome and the level of sexual dimorphism

We have generated the tear proteome of the house mouse, *M. m. musculus* and detected a total of 719 proteins at 0.01 FDR (i.e., False Discovery Rate for all peptides and proteins). First of all, we reduced our data such that only the proteins that were detected in three or more individuals were further analyzed (i.e., 457 proteins). Our aim was to identify those proteins that are sexually dimorphic (Fig. 1) and those that represent the top 5% of the most abundant proteins that may characterize the mouse tears (Fig. 2A).

PLGEM analysis of the level of sexual dimorphism revealed that 68 (14.9%) out of 457 proteins identified at 1% FDR and $p < 0.05$ were sexually dimorphic, Fig. 1. Male biased proteins included 36 (7.8%) and female biased proteins included 32 (7%) successful identifications (i.e., listed in Data S1). Thus, male-biased proteins were not more common than female-biased proteins in the tear proteome. The most significant dimorphic proteins (i.e., top 5% in Fig. 2B) included the female-biased OBP7, the male-biased MUP4, the male-only ESP1, the male-biased ESP38, and several male-biased secretoglobins (SCGB1B19, SCGB1B3, SCGB2A2—Mammaglobin, SCGB2B3, SCGB2B7). Secretoglobins are found in mammalian secretions and have important roles in the modulation of inflammation and tissue repair (*Jackson et al., 2011*), are involved in removing toxins (*Zhou et al., 2011*), and may have roles in chemical communication (*Karn & Laukaitis, 2015*). Kallikrein 1-related peptidases were also significantly sexual dimorphic (i.e., female-biased KLK1B22, KLK1B1, and KLK1B3), however, this pattern (though significant) was not consistent across all the females tested. KLKs are known as network members that are crucial for homeostasis of stratified epithelia and for activating antimicrobial Cathelicidins (*Kasparek et al., 2017*). Interestingly, we have also detected sexually dimorphic BPI proteins. Bactericidal/permeability-increasing proteins (BPI) are ∼50 kDa proteins that are a part of the innate immune system, and have an antibacterial activity against the gram-negative

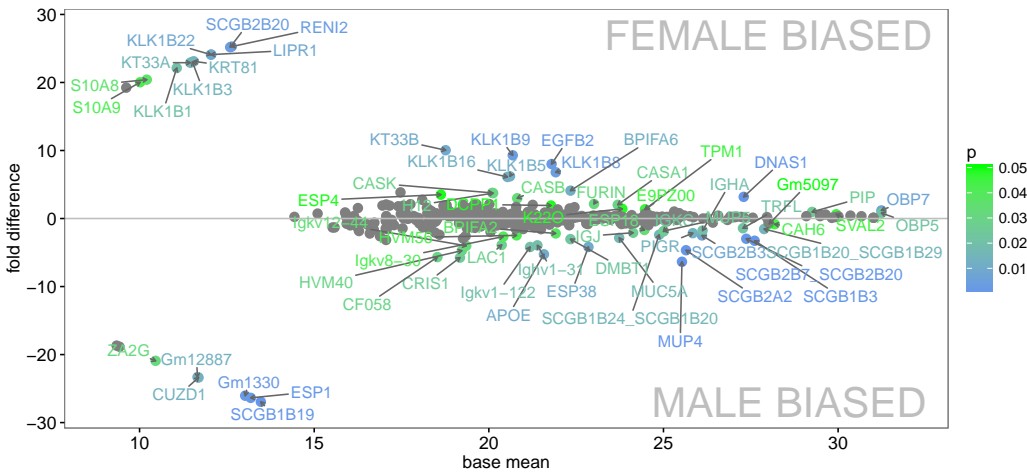

**Figure 1** **Graphical representation of protein signal intensities from LC-MS/MS (*X* axis) and particular fold differences between males and females.** Proteins are visible as mean values of the signal (base mean peak areas) and in three clusters i.e., male-unique (below the baseline - left), female-unique (above the baseline - left), and proteins present in individuals of both sex –close to the baseline ($y = 0$). Significant differences between males and females above or below the *Y* co-ordinate (fold differences) are continuously scaled from green ($p < 0.05$) to blue ($p < 0.01$).

bacteria (*LeClair, 2003b*). We have detected three BPIs, of which BPIFA2 was male biased, BPIFA6 was female biased, whilst males and females equally expressed BPIFB9B.

## The most abundant tear proteins

Based on the median value we sorted our data to detect the most abundant proteins in the tear proteome. The top 5% of the most abundant proteins that characterize the soluble tear proteome of the mouse are depicted in Fig. 2A, and include for example the female-biased lipocalins OBP5, OBP7, the unbiased lipocalins OBP1 and LCN11, and the male-biased lipocalin MUP4, Fig. 2C. Other proteins dominating the soluble tear-proteome included three male-biased secretoglobins (SCGB1B3, SCGB1B20/SCGB1B29, SCGB2B20/SCGB2B7), two unbiased secretoglobins (SCGB1B2, SCGB2B2), male-biased carbonic anhydrase 6 (CAH6), (unbiased) exocrine secreted peptide ESP6, Lacrein, and female-biased prolactin inducible protein (PIP). Interestingly, out of the top 5% most abundant proteins, ∼50% of them (i.e., 12) were significantly sexual dimorphic. Thus, even though the level of sexual dimorphism is rather low within the complete tear proteome (i.e., 15%), those few proteins that were most abundant were often the most sexually dimorphic.

## Sex-unique proteins

We provide visual representation of all proteins using MA plot, also including potentially sex-unique proteins (Fig. 1), where significant points are colored from green ($p < 0.05$) to blue ($p < 0.01$). Female-unique proteins included S10A8/S10A9 which are calcium- and zinc-binding proteins and which play important roles in the regulation of inflammatory processes and immune responses, and can induce neutrophil chemotaxis and adhesion (*Vogl et al., 2007*). We have also detected the female-unique secretoglobin SCGB2B20.

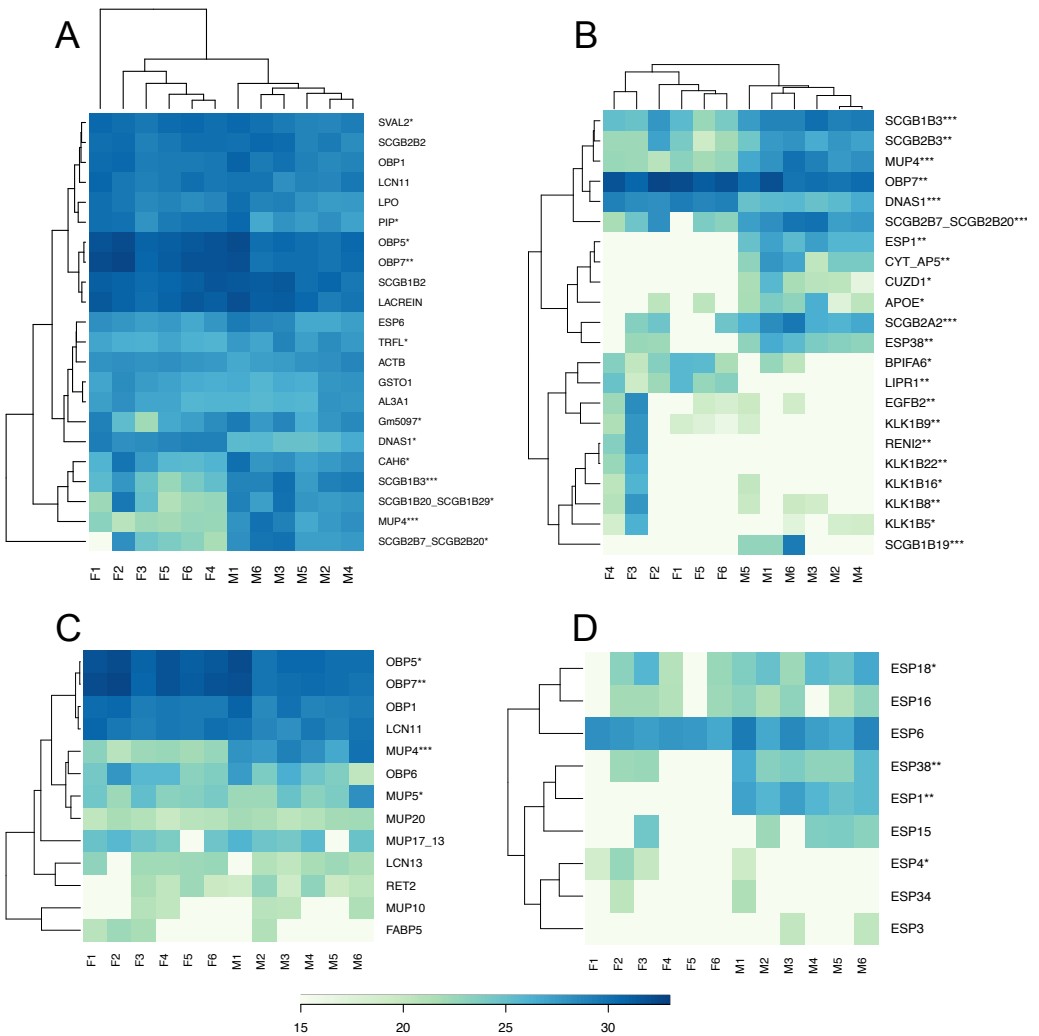

**Figure 2** **Graphical representations of individual variation in protein abundances with heat maps.**
Similarities between proteins and individuals were detected with a hierarchical clustering method: (A) the
top 5% of highly expressed proteins include OBPs, SCGBs/ABPs, and ESPs; (B) the top 5% of the most
significant sexually dimorphic proteins ($p < 0.02$) include ESPs, secretoglobins, lipocalins and female-
biased antimicrobial protein BPIFA6. There is a notable variation between individuals in lipocalin (C) and
ESP (D) abundances. Note that the expression of MUP20/Darcin is invariant over individuals. Asterisks
represent: * $P \leq 0.05$, ** $P \leq 0.01$, *** $P \leq 0.001$).

Other female-unique proteins invloved RENI2, LIPR1 and one keratin (KT33A). Male-
unique proteins included the Secretoglobin SCGB1B19, the exocrine gland-secreted peptide
ESP1, Zn-Alpha2-Glycoprotein (i.e., ZA2G), zona pellucida-like domain-containing
protein 1 (CUZD1), and products of the two predicted genes Gm12887 and Gm1330.
ESP1 was co-expressed with other ESPs (ESP3, ESP4, ESP6, ESP15, ESP16, ESP18, ESP34,
ESP38) in tears, Fig. 2D. Though male-unique in tears, ESP1 is present in male and
female saliva (*Stopka et al., 2016*). To add, various proteins that were previously detected
as sex-unique now seem to be rather sex-biased and not sex-unique/limited when new

LC-MS/MS techniques with higher detection limits are employed instead of the gel-based MS techniques (e.g., *Karn & Laukaitis, 2015*).

## Transcriptome: mRNAseq based analysis of exorbital lacrimal glands

To detect lipocalins that are excreted from the exorbital lacrimal glands, we performed the analysis of size-selected transcriptomes from ~400 bp to ~1,300 bp demonstrated with histograms in Fig. 3A. Top ten percent of highly expressed transcripts included *Scgb2a2*, *Pip*, *Lcrn*, *Scgb2b2*, *Scgb1b2*, *Scgb2b24*, *Spt1*, *Lcn11*, *Obp5*, Scgb2b7, *Esp6*, *Esp15*, *Scgb1b3*, *Sval2*, *Obp7*, *Scgb2b3*, *Bpifa2*, *Bglap3*, *Esp16*, *Scgb1b7*, *Gm20594 (Mtrnr2l)*, *Wfdc18* (underlined are the transcripts encoding proteins that were detected within the top 10% of highly expressed proteins). However, direct comparison between transcriptomic and proteomic datasets was not possible, because the transcriptome was prepared from selected ranges of gel extracted tissue mRNAs whilst the tear proteome over-represents the soluble fractions of all proteins. Next, we searched for sexually dimorphic genes that may account for sex-specific differences with the *DESeq* routine within the *Bioconductor package* (*Gentleman et al., 2004*). We have filtered for further analysis only the data where the sum of counts per row ≥10. Then, we normalised the data with a size factor vector to make the libraries comparable. Because *DESeq* calculates sexual dimorphisms from the original non-transformed number of counts we first looked at the level of variation between replicates within sex. When dispersion values are plotted against the means of the normalised counts (Fig. 3B) it is evident from the slope of the red fitting curve that data with a low mean of normalized counts have higher levels of dispersion than high expression data.

Next, we searched for differentially expressed genes by calling the *nbinomTest* in *DESeq*, vizualized in Fig. 3C. We have detected a total of 6 female-biased genes (*Obp5*, *Obp7*, *Obp8*, *Spt1*, *Hba*, and *Scgb2b1*) and a total of 17 male-biased genes (*Scgb2b7*, *Scgb1b20*, *Scgb1b3*, *Scgb1b7*, *Esp18*, *9530002B09Rik*, *Scgb2b3*, *Esp16*, *Mup4*, *Esp24*, *Cyp4a12b*, *RP23-421B1.4*, *Nop10*, *Scgb1b28-ps*, *Esp1*, *Scgb2b20*, and *Pigr*), Fig. 3D. Next we asked which of the above sex-biased genes are most differentially expressed. Using Benjamini–Hochberg corrections we have generated new 'p-adjusted' values. These genes (*p*-adjusted < 0.05) included a total of 13 genes with female-biased *Obp5*, *Obp7*, *Obp8*, and *Spt1*, whilst male-biased genes included *Mup4*, five secretoglobins (*Scgb2b7*, *Scgb1b20*, *Scgb1b3*, *Scgb1b7*, and *Scgb2b3*), two *Esp* s (*Esp16*, *Esp18*) and the gene *9530002B09Rik* (synonym: *Vpp1*—Ventral prostate predominant l, which was originally thought to be exclusively expressed in the prostate (*Wubah et al., 2002*)). The resulting pattern is plotted using MA plot (Fig. 3C) with red colouring of those genes that are significant at *p*-adjusted < 0.05 whilst all data with *p* < 0.05 (not corrected) are vizualized in Fig. 3D and those that have *p*-adjusted <0.05 are depicted with asterisks. Partial support for mRNA/protein concordance is provided in Fig. 3E, showing that those transcripts that are significantly sexually dimorphic and represent the soluble protein fraction are also detected on the level of protein. Furthermore, significant sexual dimorphisms of *Obp7* and *Mup4* (Figs. 3D, 3E) are in agreement with our proteomic analysis in this study and with our previously published study using qPCR (*Stopkova et al., 2016*). Our simple RNAseq analysis revealed the expression of *Obp8* which was previously detected only with bioinformatics tools (*Stopková et al., 2009*; *Stopkova et al., 2016*), thus providing the first evidence for the expression of *Obp8* transcript. Interestingly, OBP1 was

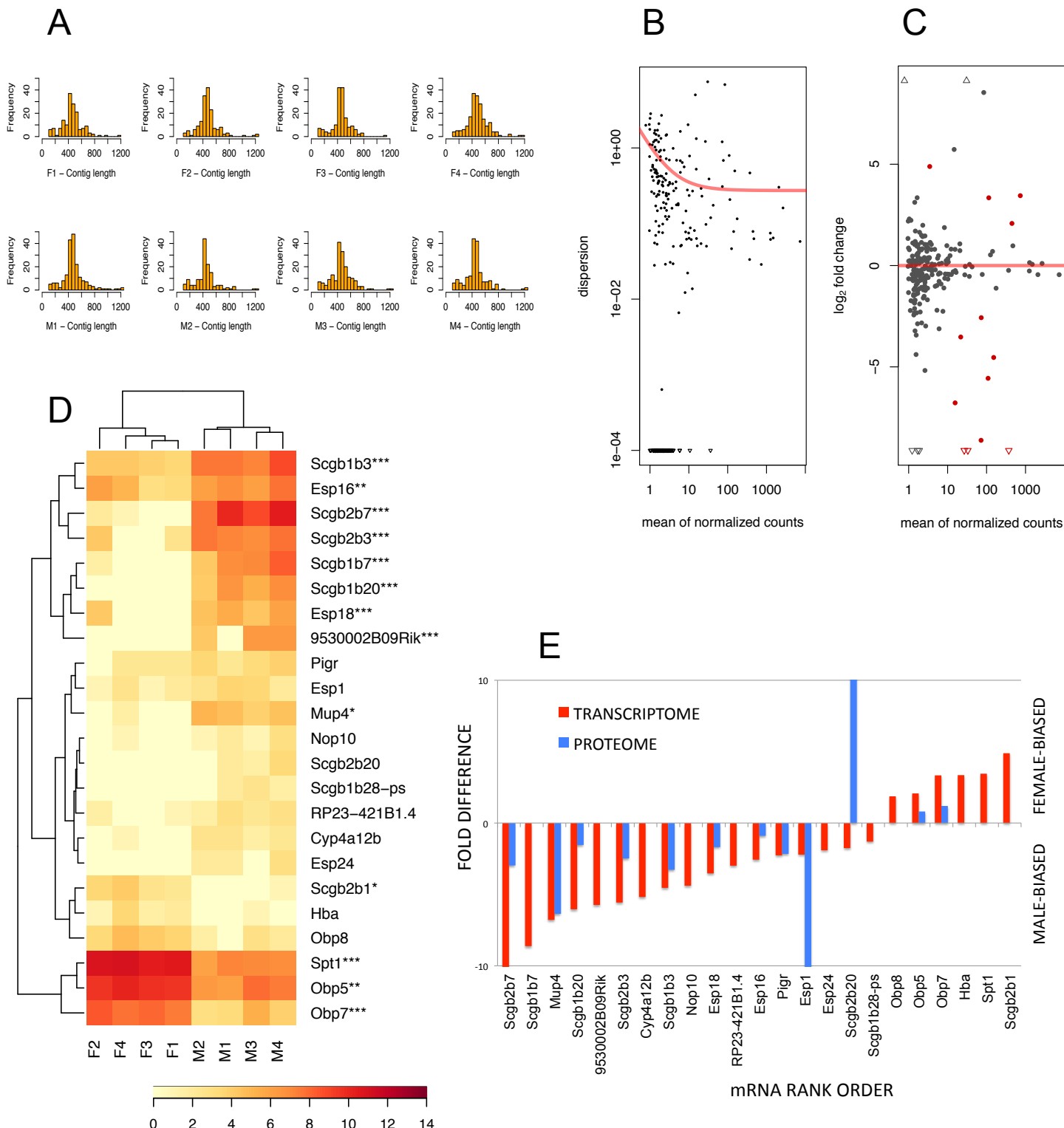

**Figure 3** **The RNA-seq analysis output.** Histograms of mRNA contig lengths from size-selected transcriptomes (A) are consistent over individuals and show that more than 50% of contigs is longer than 400 bps and not exceeding 1,300 bps. (continued on next page...)

**Figure 3 (…continued)**
Dispersion plot (B) shows the decreasing variation in signal intensities, and along with MA plot (C) are demonstrating that the transcripts with lower number of reads have a higher dispersion. Significant sexually dimorphic abundances based on $p < 0.05$ are demonstrated with the hierarchically clustered heat map in (D), with $p$-adjusted values provided with asterisks (* $P \leq 0.05$, ** $P \leq 0.01$, *** $P \leq 0.001$). (E) is demonstrating a partial support for a concordance in fold differences between mRNA expression ($p < 0.05$) and particular protein abundances for example for OBP5, OBP7, MUP4, ESPs and several secretoglobins. This relationship, however, is not linear (i.e. note huge differences in ESP1 or SCGB2B20 abundances) thus suggesting multiple sources of expression.

one of the most abundant tear proteins in individuals of both sex. However, the expression of *Obp1* transcript was medium/low in this study and low with qPCR in our previous study (*Stopkova et al., 2016*). Thus, OBP1 could be a product of several other tear-secreting glands including infra-orbital glands, accessory lacrimal glands and/or epithelial cells of ocular mucosa.

## Anti-microbial peptides

BPI proteins have an antibacterial activity against gram-negative bacteria (*LeClair, 2003b*). The saliva proteome contains seven members of the bactericidal/permeability-increasing proteins (i.e., BPI *Leclair, 2003a*; *LeClair, 2003b*) which are male biased (*Stopka et al., 2016*) and include BPIA1, BPIB1, BPIB2, BPIB3, BPIFA2, BPIFB5, BPIFB9B (*Stopka et al., 2016*). However, tears only contain BPIFA2/*Bpifa2*—one of the most expressed transcripts, female-biased BPIFA6, and un-biased BPIFB9B. Thus, we searched for other proteins/peptides which may have similar roles due to their amphipathic structural properties or proteolytic activities. Recently, WFDC proteins (i.e., 'Whey acidic proteins four disulphide core') were shown to have anti-microbial properties (*Scott, Weldon & Taggart, 2011*) and the two members WFDC12 and WFDC18 are present in mouse saliva as proteins encoded by submandibular gland transcripts (i.e., *Wfdc12*, *Wfdc18*) (*Stopka et al., 2016*). In this study, we have detected WFDC12 and WFDC18 as transcripts of the exorbital lacrimal glands (i.e., *Wfdc12*, and the highly expressed *Wfdc18*), but only WFDC18 was detected in tears on the proteomic level and just in two males. Our results, however, provide evidence that the major antimicrobial protein in tears is TRFL (Lactotransferrin), Fig. 2A. Lactotransferrin also known as lactoferrin (LF) has antimicrobial properties (bactericidal, fungicidal) and is a part of the innate immune system, mainly at mucoses (*Sanchez, Calvo & Brock, 1992*). In the tear proteome, we detected the male-biased TRFL as one of the most abundant proteins and similar amounts were previously also detected in saliva (*Stopka et al., 2016*).

## Homology modelling of representative lipocalin structures

OBPs and MUPs are likely to have complementary roles because OBPs are less hydrophobic and have higher iso-electric points whilst MUPs are more acidic and hydrophobic (*Stopkova et al., 2016*). In Fig. 4, we provide four representative lipocalin structures from homology modelling, thus showing that different lipocalins have similar structures but different electrostatics properties. The distribution of negative and positive residua is not random in OBP1 and even less so in LCN2. The structure of LCN2 is amphipathic because the interaction is driven by ionic strength between positively charged amino acid residues at the barrel opening of LCN2 and negatively charged siderophores. Thus, LCN2 is antimicrobial, as it efficiently scavenges for catecholate-type siderophores which bacteria

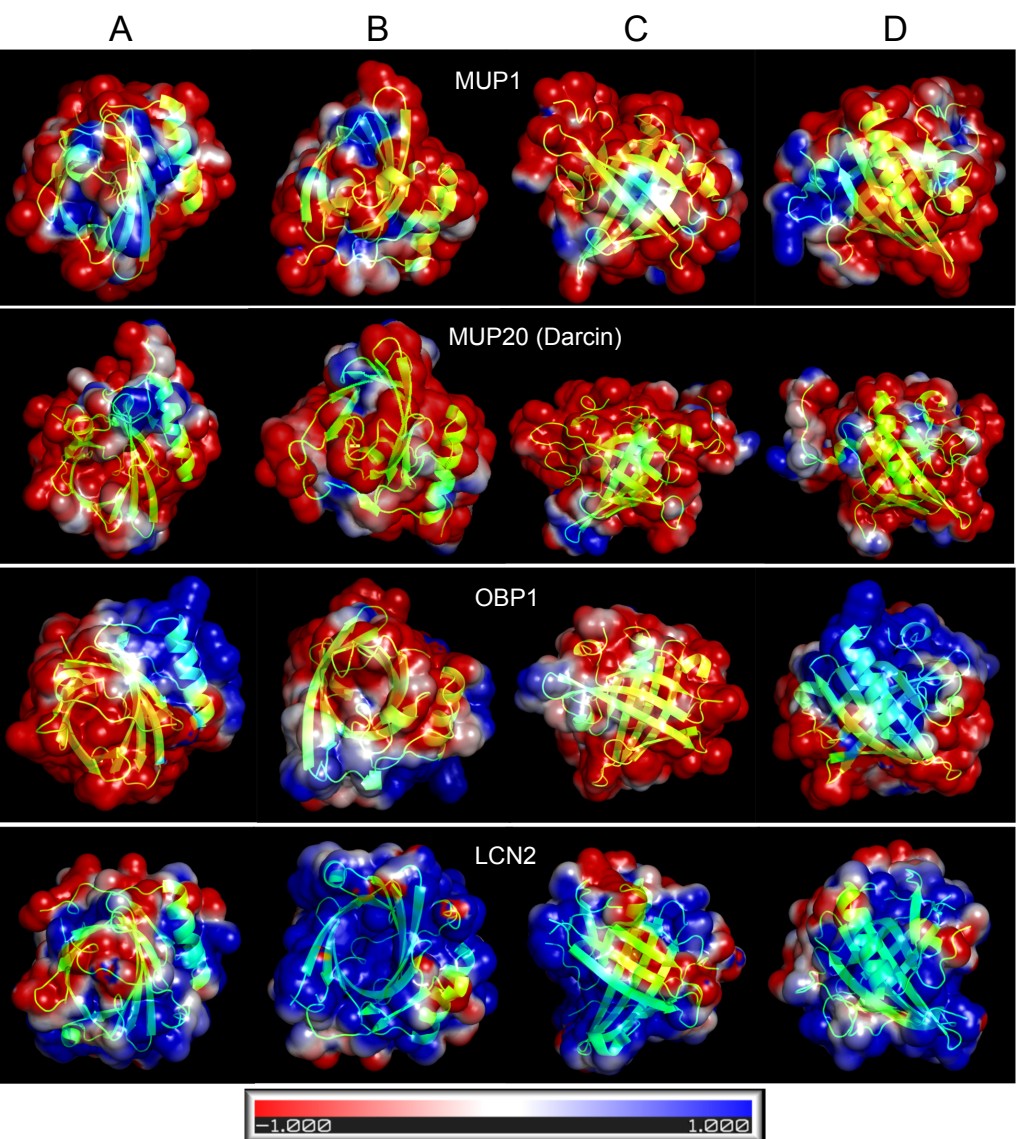

**Figure 4** **Graphical representation of the tertiary structure of MUP1, MUP20, OBP1, and LCN2 with electrostatics modelling, and scaled from -1kTe (red, negative) to +1kTe (blue, positive).** Each protein is demonstrated in the four views: lower part of the barrel (A), opening of the barrel (B), and the two side views (C, D). Although, their structures are highly similar due to their beta-barrel structures, the distribution of positive and negative charges are non-random with OBP1 and LCN2 being amphipathic. Note the positively charged amino acid residues of LCN2 at the opening of the barrel, which bind negatively charged siderophores, whilst OBP1 has most positively charged residua on its surface and alpha helix.

produce to scavenge for free iron (*Flo et al., 2004*). The structure of OBP1, however, is amphipathic due to a non-random distribution of positively charged residua on its surface and may potentially be antimicrobial (i.e., similar to CRAMP/CAMP (*Gallo et al., 1997*)). In addition, such amphipathic structure of OBP1 may also aid to a direct attack upon negatively charged bacterial membranes by its oppositely charged OBP1 surface residua including the positively charged alpha helix.

## DISCUSSION

Tears are a source of chemical signals involved in sexual signalling and are produced by sexually dimorphic lacrimal glands and their mRNAs (*Richards et al., 2006*) which code for various soluble proteins that are involved in chemical communication (*Karn & Laukaitis, 2015*; *Kimoto et al., 2005*; *Remington & Nelson, 2005*; *Sharrow, Novotny & Stone, 2003*; *Stopkova et al., 2016*). However, comparative data using label-free quantification without gel-based or Western blotting methods was to date missing. Thus, we focused on the detection of differentially abundant proteins in tears with label-free LC-MS/MS techniques to obtain more complex view on sexual signalling. We assumed that sex-specific differences in the expression of signal transporters that we detected may have roles in sexual signalling. John Maynard Smith and David Harper defined a signal as '...any act or structure which alters the behaviour of other organisms, which evolved because of that effect, and which is effective because the receiver's response has also evolved' (*Maynard Smith & Harper, 2003*). Thus, evolution of chemical communication seems to require two steps. However, the 'toxic waste hypothesis' (*Stopkova et al., 2014*; *Stopková et al., 2009*) or the theory entitled 'The origin of chemical communication by means of toxic waste perception' requires only one step because it presupposes that only the receiver's response has evolved as an adaptation to already existing sources of individual VOCs/odours which resulted from metabolic degradation. Moreover, this theory expects that the level of degradation correlates with energy intake and immune system efficiency, and thus reflects an inherent quality of the signaller.

The tear proteome of the house mouse provides a support for this hypothesis. First, the tears contain anti-microbial peptides/proteins with some of them being sexually dimorphic (e.g., BPIFA6, BPIFA2, TRFL, PIP). Thus, these proteins may yield sexually dimorphic products of bacterial degradation. Second, we have detected OBPs that are known for their capacity to scavenge for toxic substances such as 4-Hydroxynon-2-enal (HNE). HNE is a product of ocular lipid peroxidation and causes chronic inflammation (*Grolli et al., 2006*). Third, we have detected the group-A and the group-B MUPs in tears. MUPs transport pheromones and at the same time they are known to transport toxic substances out of the body (*Kwak et al., 2016*). Fourth, tear lipocalins move to the oral cavity where they were detected as proteins in the saliva where digestion starts (*Stopka et al., 2016*) including LCN3 and LCN4 which are VNO-specific (vomeronasal organ). All together, it is likely that these lipocalins may have dual functions (i.e., similarly as olfactory receptors play other roles besides the detection of chemical signals (*Ferrer et al., 2016*))—in that they are preferentially used for removing toxic substances but those that are sexually dimorphic may yield sex-specific differences which are recognized as sexual signals. Moreover, the level of sexual dimorphism in the expression of chemosensory receptors (i.e., in VNO and MOE—main olfactory epithelia) is rather limited (*Ibarra-Soria et al., 2014*). Thus, it is more likely that differential odorant detection is utilized via differential expression of chemical signal transporters.

In this study we have detected the expression of *Obp1*/OBP1, *Obp2*, *Obp6*/OBP6, and the sexually dimorphic *Obp5*/OBP5, *Obp7*/OBP7, and *Obp8* in lacrimal glands/tears. Contrary

to other *Obp*s expressed in various orofacial tissues, *Obp6* was (to date) detected only in exorbital lacrimal glands (i.e., with pyrosequencing in this study and qPCR in *Stopkova et al., 2016*). OBP5 is involved in rapid internalization of OBP-odorant complexes into lysosomes and scavenges for toxic products of free radical exposure (*Grolli et al., 2006*; *Strotmann & Breer, 2011*). However, due to the sexual dimorphism detected in this study, the presence of OBPs in tears implies their parallel roles. It is possible that all OBPs are required for the internalization of degradation products or for a transport of these harmful substances to the oral cavity where digestion starts (*Stopka et al., 2016*), whilst those that were detected as sexually dimorphic (i.e., the female-biased OBP5, OBP7, *Obp8*) may—at the same time—be essential for female sexual signalling with the products of metabolic degradation that correlate with an inherent quality of the signaller. This hypothesis, however, requires further testing.

The most interesting result of this study is evidence that males differ from females by a cocktail-like composition of significant sexually dimorphic proteins. Previously, we have demonstrated on the level of mRNA, that lacrimal glands produce high quantities of *Mup4, Lcn11, Obp5, Obp6*, and *Obp7* transcripts in the two house mouse subspecies *M. m. domesticus* and *M. m. musculus* (*Stopkova et al., 2016*). This base-line study led us to an idea that sex-specific and sex-biased expression of several different lipocalins is combinatorial, thus differentially contributing to individual scents. The combinatorial and context dependent effect of signalling has recently been described for urinary MUPs in mice (*Kaur et al., 2014*). However, in the light of new evidence, MUPs are neither polymorphic nor individually unique (*Enk et al., 2016*; *Thoss et al., 2016*; *Thoß et al., 2015*). Thus, stronger effects may be achieved by the differential expression of structurally different and sex-biased tear lipocalins with a notable variation between individuals detected in this study.

To conclude, females are characteristic of producing higher quantities of OBPs and SPT1, in tears whilst males produce more ESPs, MUPs and secretoglobins (i.e., for a comparison, see the tear and saliva proteomes of the laboratory mouse (*Blanchard et al., 2015*; *Karn & Laukaitis, 2015*)). One particular MUP - MUP20 (darcin) was surprisingly found in male and female tears and because their content is continuously moving via naso-lacrimal ducts to nasal, vomeronasal, and oral cavities where MUP20 was detected in saliva (*Stopka et al., 2016*), it is difficult to imagine that this protein functions as a protein pheromone (i.e., sensu *Roberts et al., 2010*). This is also supported by the fact that darcin is not required for sexual signalling in the laboratory mouse (*Liu et al., 2017*) and is also expressed by females in their oviductal horns and uterine liquid, thus, it is not even male-unique (*Yip et al., 2013*). Furthermore, it is possible that lipocalins (i.e., including MUP20) in the ocular tear film may have the capacity to bind air-born volatiles during social contacts and transport them to nasal tissues where they are detected as signals.

## ACKNOWLEDGEMENTS

We are very grateful to Helena Uhlířová for her careful and patient lab assistance and to Karel Harant and Pavel Talacko from the Mass Spectrometry and Proteomics Service Laboratory, Faculty of Science, Charles University in Prague for performing the LC-MS/MS run.

### Funding

This research was supported by the Czech Science Foundation (GACR, P506/12/1046), and by the project BIOCEV (CZ.1.05/1.1.00/02.0109) and National Program for Sustainability II (LQ1604) from the Ministry of Education, Youth and Sports. BK was supported from a student grant SVV 260 434 / 2017. The funders had no role in study design, data collection and analysis, decision to publish, or preparation of the manuscript.

### Grant Disclosures

The following grant information was disclosed by the authors:
Czech Science Foundation: GACR, P506/12/1046.
BIOCEV: CZ.1.05/1.1.00/02.0109.
Ministry of Education, Youth and Sports: LQ1604.
SVV: 260 434 / 2017.

### Competing Interests

The authors declare there are no competing interests.

### Author Contributions

- Romana Stopkova and Petr Klempt conceived and designed the experiments, performed the experiments, wrote the paper, reviewed drafts of the paper.
- Barbora Kuntova conceived and designed the experiments, performed the experiments, wrote the paper, prepared figures and/or tables, reviewed drafts of the paper.
- Pavel Stopka conceived and designed the experiments, performed the experiments, analyzed the data, contributed reagents/materials/analysis tools, wrote the paper, prepared figures and/or tables, reviewed drafts of the paper.

### Animal Ethics

The following information was supplied relating to ethical approvals (i.e., approving body and any reference numbers):

All animal procedures were carried out in strict accordance with the law of the Czech Republic paragraph 17 no. 246/1992 and the local ethics committee of the Faculty of Science, Charles University in Prague chaired by Dr. Stanislav Vybíral specifically approved this study in accordance with accreditation no. 27335/2013-17214 valid until 2019. Animals were sacrificed by cervical dislocation.

### DNA Deposition

The following information was supplied regarding the deposition of DNA sequences:

The transcriptome data is provided as bam files in 'Sequencing Read Archive' (www.ncbi.nlm.nih.gov/sra) under the accession numbers SRP063762 and BioProject: PRJNA295909.

### Data Availability

The raw data has been supplied as Data S1.

## Supplemental Information

Supplemental information for this article can be found online at http://dx.doi.org/10.7717/peerj.3541#supplemental-information.

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
