# Peer review of "On the tear proteome of the house mouse (Mus musculus musculus) in relation to chemical signalling"

_PeerJ, doi:10.7717/peerj.3541_

## Round 0.1 · original submission · Major Revisions

Please address all the comments and concerns of reviewers 2 and 3. Specifically:

I agree with both reviewers that the title does not represent the paper well - please rephrase it.

As discussed by both reviewers, the introduction needs to be modified in order to motivate this study - the argument for characterizing the proteome in particular is not sufficiently set up. Why was a proteome necessary, given the previous studies that were cited (e.g. the 2016 Stopkova RNA-Seq study)?

Figures - I agree that the legends need more information. Additionally, 3a isn't mentioned in the text, and Figure 4 is not explained in the Results section.

In addition to the reviewers' comments, can you please address the following:

l. 205 (RNA-Seq section) - What gel cleaning method did you use? Might be better to include the RNA prep section here instead of in section 'Sample collection'

Are sections 'Expression profile of the mouse exo-orbital lacrimal gland' (l. 205) and 'Rapid Library Preparation and GS Junior Transcriptome Sequencing' (l. 229) referring to the same sequencing experiment? Why are these two sections interrupted by the cDNA prep section? They should be merged to avoid confusion.

l. 387 (Section titled 'Supporting evidence for the theory entitled ‘The origin of chemical communication by means of toxic-waste perception’) - Please integrate this section into the Discussion.

Reviewer 1 ·

Basic reporting

no comment

Experimental design

no comment

Validity of the findings

no comment

Additional comments

This manuscript is clearly written. The authors' hypothesis was tested and sufficient supporting data were provided. Scientists will benefit from this work.

·

Basic reporting

Fail:

1. Authors investigated the tear proteome of wild house mouse by investigating tear proteins (separately from 7 individuals of each sex) using label-free LC-MS/MS techniques. They state in the abstract (& Results) that they detected 719 proteins of which ~20% displayed sex differences in their abundance. Details of all 719 proteins are not provided in the manuscript. The supplementary file lists 457 proteins and in Results section, it is stated that these 457 proteins represent a reduced number (proteins that were detected in 3 or more individuals). Why then should the authors mention 719 proteins in the Abstract, if identities of all these proteins are not provided in any form in the manuscript? In the Results, it is stated that 68 proteins having sexually dimorphic expression (36 male-biased & 32 female-biased) were identified among these 457 proteins (i.e. 14.8%). Why is it then stated in the Abstract that 20% of 719 proteins (which should be 143 proteins) were found to be sexually dimorphic? Again, where is the list of the extra 75 proteins (143 minus 68) which show sexual dimorphism? These proteins are not listed in the supplementary data, which identifies only the 68 significantly sexually dimorphic proteins in the list of 457.
Some of the identified proteins were categorized on basis of their known or proposed roles. Within the 457 proteins, the number of female-biased proteins found was almost equal to male-biased proteins. Their results showed that male tears differed from female tears in their cocktail composition of sexually dimorphic proteins, which included lipocalins (OBPs & MUPs), secretoglobins, ESPs and other proteins. Females generally had more of certain OBPs in their tears while tears of males had more of specific MUPs, secretoglobins and ESPs. mRNA seq analysis of exorbital lacrimal glands was also done (as stated in M & M). However, not much regarding findings of RNA seq is stated in Results & Discussion section. A supplementary file is provided (m RNA seq count table) and the transcriptome data is submitted to NCBI and can be accessed.
Role in sexual signaling is suggested by the authors for the male-and female-biased proteins. Authors suggest that the 11 different lipocalins (MUPs & OBPs) detected in tears with their abundances being unique for each sex and having notable variation between individuals may have a combinatorial role in chemical signaling. Since flowing tears has intimate contact with nasal mucosa (both olfactory and VNO), the authors point out that their detection of MUP20 (darcin) a male-specific urinary lipocalin, in tears of both sexes of mice, makes it difficult to imagine that this male-specific urinary lipocalin is perceived (in their nose) as a male pheromone by females as was recently demonstrated by other investigators. It is also suggested by the authors that the sex unbiased tear lipocalins (certain MUPs and OBPs) on the other hand, might serve to bind and remove toxic waste from mucosal surfaces while they flow down to the oral cavity or they may have other biological roles.

2. The title is very inappropriate. There is no real justification for it. No investigation(s) showing any hard evidence for a role of tear lipocalins in chemical signaling or toxic waste disposal is reported in this manuscript. The present title is therefore unacceptable. The toxic waste hypothesis (of Stopkova et al 2009) does not seem to be novel. That lipocalins in tears (& in other secretions and tissues) may function as a physiological scavenger of harmful waste compounds to protect body cells (e.g. of ocular surface and other tissues) has been proposed earlier several times in different publications (both without and with, some accompanying experimental evidence). Several examples are listed below:
i) Introduction & Discussion of BBA (2005), 1729, 154-165, and also references therein [for human tear lipocalin LCN1 (a non-OBP lipocalin) and for OBP-like lipocalins of hamsters (FLP & MSP) which are female-specifically expressed in lacrimal glands & secreted abundantly in tears]; ii) JBC (2007), 282, 31493-31503 (for lipocalin alpha1 microglobulin); iii) J Cellular Physiol. (2005), 202, 683-689 (for lipocalin Ex-FABP); iv) FEBS Journal (2006), 273, 5131-42 (for bovine & pig OBPs); v) J Histochem. Cytochem. (2002), 50, 433-5 (for LCN1/pituitary); vi) Biochemistry (2002), 41, 11786-94 (elephant trunk OBP); vii) Lab Chip (2008) 8, 678-88 (rat OBP-1F).
Regarding the first part of the title (“roles of tear lipocalins as chemical signaling..”), this is also not a new proposal……it has been suggested long back that certain tear proteins of hamster (OBP-like lipocalins as well as non-lipocalins), which are abundantly expressed in lacrimal glands and secreted in tears in a sex-specific/-biased manner, may function in chemical signaling [BBRC (1995) 208, 412-417; J Steroid Biochem Molec Biol (1999) 70, 151-8; BBA (2005) 1729, 154-165; Gen Comp Endocrinol (2008) 158, 268-276; BBA (2007) 1771, 55-65; BBRC (2006) 341, 1286-93]. See also 3 abstracts (I-7, I-11 & III-5) of presentations at Benzon Symposium no:50 (2003) and other publications by the same authors, for a chemical signaling role proposed for a male-specific salivary lipocalin abundantly expressed in submandibular gland of hamsters.

Relevant prior literature is not appropriately cited:
Surprisingly, there is no mention in the Introduction or Discussion of the sex-specific and sex-biased tear (and salivary) OBP-like lipocalins (and other proteins) identified in hamster, which have been extensively characterized and reported earlier. This lapse needs to be rectified.

3. The quality and clarity of the English language used in Introduction, Discussion and in some portions of Results is not satisfactory. Language needs to be considerably improved. In the present form, the manuscript is quite tiresome to read and comprehend. The authors need to work on this and maybe take professional help. However, the language in the Material & Methods section and some parts of Results is OK. The Introduction is very long and unfocused. It should be shortened. The authors have tried to cram in lots of information in the Inroduction on MUPs, OBPs, ESPs, Secretoglobins…some of these are not essential to build a case for their investigations.

4. In the Introduction, Results and Discussion sections, the authors deviate (interrupt) frequently to describe or discuss at length, their findings in other recent publications (e.g. Stopkova et al 2016, Stopka et al 2016, Stopkova et al 2014). This causes confusion for the reader (examples: the sentence in Discussion, starting in line 490 and ending in line 495; the sentence starting in line 445 and ending in line 453; sentence starting line 436 and ending in 439). These detailed descriptions/information are not very helpful and could be removed or made succinct.

5. Legends for Figs 1, 2, 3 & 4 are too brief. They authors should elaborate them to be more understandable by the unfamiliar reader. What does Y axis (base mean) represent in the MA plot in Fig 1? What is “dispersion” in Fig 3? No legend, explanatory note is provided along with the supplementary raw data file. This is required for the unfamiliar reader to comprehend all aspects of the data provided.

Experimental design

1) Experimental design seems OK. Material & Methods section is sufficiently detailed. Authors investigated proteins present in ocular surface washings (“eye lavage”; lines 144-146). In addition to proteins secreted by exorbital lacrimal gland, the ocular surface washings (eye lavage) would contain proteins secreted by the infra-orbital lacrimal gland, accessory lacrimal glands (Harderian and Meibomian glands of eye lids) and also proteins secreted out by epithelial cells of ocular mucosa. This needs to be mentioned at an appropriate place within the manuscript.
It should be made clear why protein data from “tears” 6 male and 6 female mice are presented in the manuscript when it is stated in M & M that tears were collected from 7 mice of each sex (i.e. 14 individuals; line 136-139). Were the ocular surface washings (containing diluted tears) taken from each eye of a mice and then pooled? This should be clarified in M & M. When the exorbital lacrimal glands were excised for RNA isolation (after sacrifice on the day after eye lavage) were a pair of glands from each animal pooled to make a tissue sample? It should also be clarified why “sequencing was conducted on whole lacrimal gland of four wild caught adult female and four adult male biological replicates”? (line207-208) Why not from seven male and seven female glands from which RNA was isolated?

Validity of the findings

OK, but some comments are given below.
The title cannot be a speculation.

Drawing similarities of OBPs and LCN2 with the antimicrobial peptides (similar to CRAMP) on basis of their common amphipathic nature is untenable in the absence of any structural or sequence similarity. LCN2 reportedly possesses antimicrobial activity but then it is due to a different mechanism depriving bacteria of iron essential for survival (by scavenging of bacterial siderophores). CRAMP peptides are antimicrobial due to their ability to puncture holes in membranes.

Additional comments

1. Lines 500-505/506: Similar bioinformatics analysis (i.e. lipocalins of different species having CXXXC protein motif were predicted to be located on X-chromosome) was first performed much earlier and presented in Benzon Symposium no. 50 (2003) (abstract no: I-7). This analysis involved, OBPs and OBP-like proteins and probasins of mouse, rat and hamster and included in addition to aphrodisin, the sex-specific lacrimal and salivary gland lipocalins of hamster (FLP & MSP) which are secreted in tears and saliva.
Authors should consider whether the lines 500-505/6 are to be retained. What is the relevance of it in context of the work reported in the manuscript?

2. There are some mis-statements: line 504-506. Please note that no ligands of the lipocalin aphrodisin has been identified, which has been shown to be responsible for eliciting copulatory behavior in male hamsters (independently or in association with the lipocalin protein). The lipocalin purified from vaginal discharge and devoid of any detectable ligand is shown to be effective as a pheromone but requires contact with tongue or nose (accomplished likely by licking and snout contact) (J Biol Chem. (1986) 261, 13323-6); however, in a study performed later, the recombinant version of the protein did not show appreciable aphrodisiac activity.

Some language related suggestions:

In the abstract, line 13, why “However..”? In line 12, “may function” instead of “to function” is more appropriate. Consider replacing “are” in line 17 and “demonstrating” in line 22 with less assertive words (something like “may”, which is used later in line 17). Again in line 32, “able” should be changed to “potential”: Only some lipocalins (some urinary MUPs of mouse and nasal OBPs of pig and cow) have been shown to bind and transport specific VOCs; natural VOC ligands have not been identified for many lipocalins and there is no surety that they exist (the lipocalin protein might have other yet unknown functions). Line 521, “function” could be “may function”. The sentence starting in line 521 and ending in 523 is again restating an objective and that too, in the final conclusion paragraph…this should be deleted. Line 23, “non-dimorphic” should be replaced by “non-sexually dimorphic” or “non-sex dimorphic” or “…show no sexual dimorphism in levels”. In line 38, insert, “(MUPs)” after “proteins” and shift the citation within the brackets to the end of the sentence in the same line (i.e. after ..”liver”) and start the next sentence with MUPs instead of “They”. Line 66, why “For example..”? Line 102, insert “the” after “keep”. Line 103, replace “are proteins” by “ proteins are”. Some sentences are extremely long and difficult to comprehend, e.g. the sentence starting in line 51 which extends till line 59!
These are some examples but there are more which are not pointed out.

Reviewer 3 ·

Basic reporting

-- I thought the title of the paper was misleading. The title would lead the reader to believe this was a study on the dual functional role of lipocalins: as pheromones and as a waste disposal system. However, the data in the study really are an unbiased screen of the tear proteome in wild mouse with emphasis on sexual dimporhism, and includes all kinds of interesting information about several protein families--not just lipocalins. Moreover, there are no functional studies to determine how lipocalins can have this dual role.

-- English and grammar could be improved. For instance, beginning a paragraph with "For instance" is unusual in English and defies the notion of the topic sentence.

-- Reporting on the figures was incomplete. There was incomplete referencing of all figure panels in the text (e.g., Fig. 3a is not referenced in the text). The legend for Figure 3 is incomplete. Finally, Figure 4 was referenced in the Discussion, which is quite unconventional; it was not clear why this was the case and raises questions about the novelty of the data.

-- Some points were highly speculative. For example, on line 81 the authors speculate that Mup4 could explain previous observations that mice preferentially investigate mouth and facial areas. This is speculative because there are thousands of other compounds that could explain this result.

Experimental design

1a) In the Abstract and the Introduction the objective of the research was unclear. The Introduction discusses a variety of topics: lipocalins in the mouse genome and their potential function as pheromones; excretion patterns of lipocalins; ESP excretion patterns and potential functions, along with strain differences; and the immunogenic function of a variety of proteins excreted in tears. The Introduction then concludes with the Aim of the paper on line 122, but it's not clear how all the previous material fits into this aim.
1b) Related to this first point, it seems there are already a few papers in the literature dealing with the tear proteome: two papers from Karn and Laukaitis, and the authors themselves have a 2016 paper on the tear transcriptome. Thus, it would make sense to have a piece of the Introduction devoted to saying how previous unbiased assessments of the gene products of the tear glands have contributed to the field, and how this work will build upon that. However, I found no such discussion of the state of the art of the tear proteome in the scientific literature, and how the techniques and animal subjects in this work build upon that.

2a) I also have serious concerns about the concept of pheromone in this paper. In the abstract and in several points throughout the paper, the authors suggest that, in order for a compound to be classified as a pheromone, it needs to be "sex-specific" or "sexually dimorphic." This point of view is unfounded; there are known pheromones that are not sexually dimorphic. That said, "sex pheromones" are a subclass of pheromones involved in sexual behavior, and clearly many sex pheromones exhibit sexually dimorphic expression patterns. But even in the specific case of sex pheromones there doesn't seem to be a requirement for sexually dimorphic pheromone expression, because pheromone receptors can also be dimorphic; receptor dimorphism could therefor be the point at which sex-specificty is expressed.
2b) Related to 2a, I also have concerns that the authors seem to suggest that a lack of sexual dimorphism is an indication that a protein is involved in toxic waste disposal. Could it not it be the case that a dimorphic protein is also involved in waste disposal? Indeed, the authors discover--in this paper--that BPIs are sexually dimorphically expressed. While very interesting, I think this result and the overall model need to be ironed out for logical consistency.

Validity of the findings

-- I thought the data were really interesting. This tear proteome and transcritome of Mus musculus musculus will be a valuable resource for people in the field. Thus, I think this paper would be better suited to be simply presented as survey of the musculus proteome with special attention to sexual dimorphism.

-- Regarding the transcriptome, it is conventional to follow up on top candidates with some kind of validation. I would ask that at least some of the hits be validated wtih qPCR or pyrosequencing.

---

## Round 0.2 · Minor Revisions

It is clear that the authors' intent with this paper is to provide evidence that may support their hypothesis on the evolution of chemical communication. However, both reviewers' comments imply that the focus of the introduction and discussion has shifted too far from the data at hand - the tear proteome itself. The reviewers have several suggestions on how to provide the reader with sufficient background on the biology of the lacrimal gland, as well as previous studies on gene expression in mouse lacrimal glands - please follow their advice.

Existing sections in the introduction will have to be shortened, eliminated, or moved to the discussion in order to accommodate the new material - I suggest to start with the sections on MUPs (l. 57-71), ESPs (119-126), and BPs (l. 89-95). Please also attempt to tighten the Results and Discussion sections as much as possible.

I find the phrasing of the title a bit odd - I would suggest replacing this with "On the tear proteome of the house mouse (Mus musculus musculus) in relation to chemical signalling".

I agree with Reviewer 2 that the Supplement needs legends and/or a data dictionary to fully describe the column and row labels.

Minor edits:

l. 177-180: The number of mice caught adds up to 14, not 12.
l. 301, 302 - The references to figures 1 and 2a need parentheses.
l. 473 - Please define VNO.
l. 478 - Please define OE.

·

Basic reporting

The Introduction is still very long, rambling and quite unfocused. Just because the journal is online it should not be assumed that it does not matter how long the Introduction is! The Introduction contains much unnecessary information, describes/discusses too many things with abrupt deviations which makes it difficult to stay focused and keep reading. This issue might turn off a casual reader and even a serious reader.

Description of results should be systematic and figure based with little or no extraneous talk or discussions. Reading through the results section is very difficult and irritating.

There is no legend/explanations provided for the peerj-15486-S1_Dataset. It is left to the reviewer/reader to decipher everything. Why is Fig 3 (A, B & C) required? This information can be briefly mentioned in M & M.

Experimental design

No comments

Validity of the findings

The authors have generated the tear proteome of the house mouse and identified both male-biased and female-biased proteins as well as unbiased proteins. Female-biased proteins were detected in almost the same frequency as male-biased proteins. Authors show evidence that female tears contain more of lipocalin OBPs while male tears contain more of lipocalin MUPs. ESPs & secretoglobins. Proteins displaying most sexual dimorphism in abundance belonged to the MUP, OBP, ESP & secretoglobin families. Certain lipocalins (MUP20 & OBP6) were not sexually dimorphic in tears. Sexually dimorphic bactericidal proteins were also detected in tears. RNA seq is also performed on male and female lacrimal glands of house mouse.

Authors could consider the following possibility to explain the non-sex-dimorphic presence of MUP20 in tears. It can be suggested that MUP20 (free of any ligand or containing a displaceable ligand) in tears of both males and females is essential for perception of pheromonal ligands by nose. After all both males and females need to finally perceive the volatile pheromonal ligands emanating from voided urine of another male mouse (the signaling mouse). MUP20 dissolved in tear film on the ocular surface (of the receiving mice) could trap and bind airborne volatile ligands released from voided urine of the signaling male mice and then present the ligand to the nasal receptors. This is possible since tears (containing MUP20) drains through the nasal cavity and should contact the nasal mucosa. The MUP-20 would thus act as a ferry for carrying the volatile ligand across the aqueous nasal mucus and then release it for it to contact the nasal receptors. The free ligand, which is hydrophobic would be poorly soluble in the aqueous nasal mucus and thus unable to contact the nasal receptors. The volatile ligand might be SBT or the any other volatile ligands carried by MUP20 within its hydrophobic cavity when it is present in male-mouse urine.

However, there is no attempt at co-relating/cross-checking the results obtained here in this manuscript (with respect to tear proteome of males and females & RNA seq results of male and female lacrimal glands) with results earlier obtained by others on lacrimal glands of mouse. This is still important even if the mice used earlier was different from wild mouse, e.g, domesticated laboratory mouse like BALB/C or C57BL. Were the genes, found to have marked sex-dimorphic expression in the earlier studies, also detected in the present RNA seq results? and did they also display similar sexual dimorphism?? For e.g. was the mouse lacrimal secretory protein SMARP or its transcript (reported to be highly sex-dimorphic in earlier studies; see item 2, below) also detected by the authors of the present manuscript? If so, what was the dimorphism noted? (male- or female-biased & what was the fold-difference?). Also see refs 1 & 3. RNA seq results need to be compared with the results obtained in 1, 2 & 3. It is surprising and also unacceptable that such reports on mice (some listed below) have been ignored and neither cited nor comparisons made with them in the present manuscript!

1) Influence of sex on gene expression in the mouse lacrimal gland. Exp Eye Res. 2006 Jan;82(1):13-23. Epub 2005 Jun 24. PubMed PMID: 15979613. (Richards et al)

2) A novel mouse protein differentially regulated by androgens in the submandibular and lacrimal glands. Arch Oral Biol. 2007 Jun;52(6):507-17. Epub 2006 Dec 15. PubMed PMID: 17174266. (Sakulsak et al)

3) Gender-related differences in gene expression of the lacrimal gland. Adv Exp Med Biol. 2002;506(Pt A):121-7. PubMed PMID: 12613898. (Richards et al)

4) Sexually dimorphic expression of androgen-binding protein mRNA in mouse lacrimal glands. Invest Ophthalmol Vis Sci. 2005 Jan;46(1):31-8. PubMed PMID: 15623751. (Remington et al)

5) Diverse spatial, temporal, and sexual expression of recently duplicated androgen-binding protein genes in Mus musculus. BMC Evol Biol. 2005 Jul 14;5:40. PubMed PMID: 16018816; PubMed Central PMCID: PMC1187883. (Laukaitis et al)

6) Sex-related effect on gene expression in the mouse meibomian gland. Curr Eye Res.
2006 Feb;31(2):119-28. Erratum in: Curr Eye Res. 2006 Jul-Aug;31(7-8):691. PubMed
PMID: 16500762. (Richards et al)
(Meibomian glands are accessory lacrimal glands whose secretions contain variety of lipid components and also proteins which mix with tears. Sampled tears in the present study under review should contain lipids and proteins secreted also by Meibomian gland. Additionally, certain Meibomian secreted lipids, which are tear components are suspected to be pheromones, that could be bound by OBPs/MUPs in tears.

There is also no mention in the manuscript that lacrimal gland secretory deficiency results in the debilitating condition of keratoconjunctivitis sicca (dry eyes) and that this condition is known to be much more prevalent in women than men (~9:1 prevalence ratio). The introduction could have dwelt partly on this well-known issue in humans and partly on the possibility of chemical communication by lacrimal gland secreted sex-dimorphic components. This would have helped to systematically build a case for examining the proteome of male and female tears of (wild) mouse.

Additional comments

Nothing substantial is reported in this badly written manuscript however it seems that lacking potential impact or lacking novelty, with not much degree of advance and possible readership are not a concern for this journal.

Reviewer 3 ·

Basic reporting

Title/Abstract/Introduction
These sections are now a lot more coherent and easier to follow.

For a study that is strictly an unbiased survey of the proteome and transcriptome of lacrimal glands, there is a surprising amount of text on topics that are not directly researched in this manuscript. For example, quite a lot of text is devoted to the scavenging of volatile organic compounds (VOC), especially those that might come from immune defense. However, VOCs were not analyzed here.

Similarly, I thought there was surprisingly little on the subject at hand: the tear proteome. For example, what is the biology of the lacrimal gland? Why is it critical that we have an unbiased survey of its proteome? The great thing about an unbiased proteomic screen is that it opens up new questions about the function of a tissue, and indeed there appear to be new proteins on the scene as a result of this work. However, it is clear that the researchers are trying to understand the lacrimal gland in terms of its possible role in chemical communication, and their confirmation that proteins are sexually dimorphic in these glands provides a basis for how that might work. Related to this last point, though, it’s worth mentioning that although sexual dimorphism does seem to be a prerequisite for these proteins to serve as pheromones, sexual dimorphism in gene expression is rampant. For example, this paper (http://genome.cshlp.org/content/16/8/995.full.pdf+html) shows that 72% of genes in the liver are sexually dimorphic.

Methods
These are much improved.
What is the clustering method used in the heatmaps?

Results
I appreciate that a direct comparison between transcriptomics and proteomic data is difficult because each set of data is biased—but to disregard any sort of formal comparison between the two seems a little extreme. Besides, in the introduction it is stated that the aim of this paper is to compare the tear proteome with your qPCR data from 2016; one could argue that the RNAseq data is more comparable to the protemome than the qPCR data.
Thus, I think it would be worthwhile to discuss or show the concordance between the two datasets. Just looking at the figures shows some interesting overlap, which indeed suggests that there is at least some concordance, in spite of the methodological differences.
Figure 2: The use of asterisks is very confusing, and seems to change from one panel to the next. Also, the same approach to labeling statistical significance should also be appled to the heatmap in Figure 3D.

Discussion
This is much easier to read now.

Experimental design

No comment.

Validity of the findings

no comment

---

## Round 0.3 · accepted · Accept

The reviewers' comments have been addressed sufficiently - thank you for your work on this.